# Health Impact Assessments of Health Sector Proposals: An Audit and Narrative Synthesis

**DOI:** 10.3390/ijerph182111466

**Published:** 2021-10-31

**Authors:** Nelius Wanjiku Wanjohi, Reema Harrison, Ben Harris-Roxas

**Affiliations:** 1School of Public Health & Community Medicine, University of New South Wales (UNSW), Sydney 2052, Australia; 2Centre for Health Systems and Safety Research, Australian Institute of Health Innovation, Faculty of Medicine, Health and Human Science, Macquarie University, Sydney 2109, Australia; reema.harrison@mq.edu.au; 3Centre for Primary Health Care and Equity, University of New South Wales (UNSW), Sydney 2052, Australia; b.harris-roxas@unsw.edu.au; 4Primary, Integrated and “Community” Health, South Eastern Sydney Local Health District, Darlinghurst 2010, Australia

**Keywords:** health impact assessment, health care, health services, planning

## Abstract

Background: Health impact assessment (HIA) is a tool used to assess the potential health impacts of proposed projects, programs, and policies. The extent of the use of HIAs conducted on health sector proposals, and what they focus on, is currently largely undocumented. This paper reviews HIAs conducted on health sector proposals, their characteristics and describes the settings in which they were conducted. Methods: A systematic review was conducted, including peer-reviewed journals and grey literature utilizing keywords, synonyms, and subject headings relevant to HIA and the health sector. Eligibility criteria were independently applied to the identified works and data appraisal conducted using the Critical Appraisal Skills Programme qualitative checklist tool. Results: 19 HIAs were identified and included in the review, including 13 rapid, three intermediate, and two comprehensive HIAs. The HIAs use was evident across a range of health service contexts, though all but one had been conducted in developed countries. Conclusion: The use of HIAs in the health sector is limited. There were various benefits attributed to the HIAs analysed including and not limited to the allocation of resources, reducing inequalities, and identification of possible negative consequences of a project. There is an opportunity to improve the use and reporting of HIAs across health settings internationally to enhance the consideration of broader determinants of health, influence decision making, and use of evidence in health sector planning for the future.

## 1. Introduction

Health Impact Assessments (HIAs) draw on the World Health Organization (WHO) basic principles for defining health [1] to consider a broad range of determinants of health and wellbeing. It is a structured process used by public health professionals and decision-makers to identify and communicate the potential health impacts of proposed projects, policies, and programs [2,3]. HIAs implement a broad approach to defining health, unlike the historic narrow concept where health encompasses the physical well-being from a medical perspective [4]. They provide decision-makers with recommendations around how to both maximize the health benefits and mitigate adverse health impacts of proposed projects, policies, and programs [5].

Over the past decade, HIAs have been increasingly used to inform decisions and influence the development of policies in the non-health sector [6,7]. While the use of HIA in non-health sectors such as land-use planning, transportation, energy project, extractive industries, and agriculture has expanded in recent years, the use of HIAs in the health sector has remained comparatively limited [8]. Multiple determinants of health at individual and population levels lie outside the control of the health sector and as such, public health professionals have emphasized the use of HIA to influence policies outside the health sector to enhance population health [9]. The conduct of HIAs is identified in the context of promoting equity and reducing health disparities in the community [10,11]. As such, researchers, policy maker and health practitioners should have a shared understanding of health and health inequalities [12]. HIA tool is also an approach to protect the vulnerable in the community experiencing health inequalities about sex, age, ethnicity, social, and economic status [13]. These problems would be resolved using an all-inclusive care system to avoid possible conflicts of interest in the market. 

It remains unclear why HIAs have not been widely conducted in the health sector compared to other sectors. Supposedly, there is an assumption that all health sector proposals, policies, and projects seek to enhance human health. Harris (2005) contends that health sector actions are often simply assumed to be appropriately addressing the needs of the community. Therefore, or so the argument goes, the potential risk of negative health impacts occurring as a result of health sector-led plans is minimal [14]. Real-world experience of health sector proposals contradicts this belief about the benevolence of all health sector proposals and instead highlights that health sector proposals can lead to (sometimes unforeseen) adverse health outcomes. For instance, human activities such as deforestation, irrigation projects, and dams which are aimed to increase food production and provision of clean energy can result to transformations that can adversely affect human health. A case in point is increase in prevalence and occurrence of diseases such as schistosomiasis [15].

Internationally, public health professionals continue to promote the need for increased assessment of the potential health impacts of health and other sector-based policy proposals and plans [16]. Lifsey argues that there is a growing need for evidence-based public health practices and decision-making in all sectors, including the health sector [17]. Similarly, Bos argues that increasing the number of HIAs conducted in the health sector would allow the sector to better anticipate evolving and emerging health issues and aid in the development of relevant responses [18]. To understand how HIAs may contribute to and be promoted for use on health sector proposals in the future, it is important to first establish the scope of HIAs currently being used in the sector. To date, there has been no synthesis of empirical information on the extent and nature of health sector HIAs. This paper will provide a systematic review of the health impact assessments that have been conducted in the health sector and explore the opportunities to incorporate HIAs into the planning and implementation of health sector activities.

## 2. Review Aims

To identify evidence of health impact assessment reports (identified in peer-reviewed and grey literature) conducted in the health sector to better understand the current use of HIAs in health settings. 

## 3. Methods

The Preferred Reporting Items for Systematic reviews and Meta-Analysis Protocol (PRISMA-P) were used to guide the conduct of the review and to report on the identification of HIAs in the peer-reviewed and grey literature. PRISMA-P provides a set of items to guide the conduct of a thorough review process [19]. According to Page 2021, PRISMA-P ensures transparency and completeness [20] in reporting to improve the quality of the review [21].

## 4. Eligibility Criteria

### 4.1. Inclusion

A comprehensive search utilising the PICO (Population, Intervention, Comparison, and Outcomes) tool relevant to qualitative research was integrated to provide a true representation of the available research and prevent bias [22]. The inclusion criteria for eligible HIAs were guided by and assessed to include:

Types of publications: Publications considered had to be in English and full text. No limitations or specifications were applied for the year of publication.

Intervention: Health impact assessment conducted in the health sector in any location globally.

Outcomes: Data regarding the process or findings of HIAs.

### 4.2. Exclusion

Publications that did not meet the above inclusion criteria were excluded. Publications that focused on public health, health equity, or health inequalities in broad terms but made no reference to HIA or its use were also excluded. Neither commentary nor opinion materials were included in the study. The material resources that had no direct link to HIA were excluded from the review. 

## 5. Data Sources

### 5.1. Electronic Databases

Keywords, synonyms, and Medical Subject Heading Terms (MESH terms) including and not limited to, health service, health planning and delivery of care were developed around the concepts of ‘health impact assessment’ and the ‘health sector’. MESH terms are specialized controlled vocabulary that are created and frequently updated by the National Library of Medicine (NLM) for the purpose of indexing journal, searching and cataloging for articles [23]. Four electronic databases (Medline, Scopus, Web of Science, and Global Health) were searched, in July 2021, using keywords, synonyms, and MESH terms (Appendix A). These databases were selected to ensure a wide range of international publications from a range of relevant disciplines. Publications were obtained and references were merged and assessed using the Endnote X8 [24] and X9 [25] program, where the exclusion criteria were applied, and any duplicates references were identified. The resources adapted were all not older than twenty years since the date of publication. The data search was based on 2003 as the base year and the current year (2021) as the upper limit of the study. 

### 5.2. HIA Websites

HIA websites were identified through a google search to ensure there were ways through which data about the same would be collected from the online sites. To supplement the databases and to ensure that a wide range of relevant publications were included, the website that is highly recognized as a source of HIA reports was analysed. Website repositories which are relevant to HIA, such as HIA Gateway [26], HIA Connect [27], the U.S. Health Impact Project [28], WHO [29], IAIA [30], and SOPHIA [31] were systematically searched for reports published by non-governmental organizations, governments, the private sector, and other professional organizations. Appendix A lists the websites that were searched. The website references were also managed through referencing software Endnote X8 and X9 [24,25].

### 5.3. Study Selection and Data Extraction

The first author screened all titles and abstracts according to the inclusion and exclusion criteria. Full-text documents were then obtained from publications that met inclusion criteria and all three authors independently assessed these against the eligibility criteria. A data extraction table with the following categories was developed to summarize and identify the characteristics of the included documents: author, year, country and nature, type of HIA, method, objective, outcome, and impact on decision-making and implementation, and source. 

## 6. Analytic Approach

A narrative synthesis was conducted on the included studies to examine the HIAs conducted in the health sector. The narrative synthesis relies on the utilization of words and text to summarize and provide an explanation of the findings [32]. This approach was selected as it allows consideration of a broad overview of studies (in this case HIAs) and their settings, thus enabling review of data across disciplines, and not only describing similarities and differences but also key dimensions promoting comparisons [33]. A narrative synthesis allows heterogeneous studies to be summarized and not only those focusing on a particular intervention. A data extraction table was used to summarize the relevant publications and to determine the major characteristics, concepts, and findings. This enabled identification of more than one overarching narrative in HIAs conducted on health sector proposals.

## 7. Appraisal of HIA Report Quality

The Critical Appraisal Skills Programme qualitative checklist tool (comprised of ten questions addressing issues of result validity, the actual results, and relevance to the local community) was used to assess and determine the quality and validity of reporting HIAs in the health sector amongst the included HIAs [34]. Several reviewers had been adopted in the study. Two reviewers rated each study independently and conflicts were resolved after detailed discussions between reviewers.

## 8. Results

### 8.1. Included HIAs

There were 825 records identified from the four databases, with a total of 698 records remaining once duplicates were removed. A further 688 records were deemed irrelevant to the study and excluded following title and abstract review. In addition, one on the records was not in English, therefore excluded. The full texts of nine remaining articles were downloaded and identified as eligible for inclusion. Ten additional HIAs records were obtained from the grey literature search that all met the inclusion criteria and were included in the review (see Figure 1).

### 8.2. Characteristics of Included Health Impact Assessments

Nineteen publications met the inclusion criteria and were included in the review. Key characteristics of these nineteen publications including country, nature, and level of the HIA, and the data collection methods used are summarized in Table 1. The levels of HIAs were categorized using *Health Impact Assessment: A Practical Guide* into the desktop, rapid, intermediate, or comprehensive [33].

### 8.3. Use of HIA to Assess Health Sector Proposals

There was a total of 19 HIAs identified from high- and low-income countries globally. These were limited to four high income countries and one low-income country as shown in Figure 2 below.

### 8.4. Types of HIAs

The review of HIAs found desktop, rapid, intermediate, and comprehensive HIAs have been used in a range of projects, plans, and policies in the health sector (see Table 1). In some cases, more than one type of HIA was conducted simultaneously. The type of HIA conducted was dependent on the resources available e.g., rapid and desktop [37,39,53] to ensure the amount of data collected in every stage would be a point of reference in the specific levels of communication. Thirteen of the HIAs identified in this study adopted a rapid approach [36,37,40,44,45,46,47,48,49,50,51,52], three adopted an intermediate approach [35,41,43] and two used the comprehensive HIA approach [38,42]. Four of the HIAs applied a desktop and rapid approach simultaneously [37,39,52,53]. Rapid and desktop HIAs are more cost-friendly and require less time to conduct. At the same time, the HIAs are time-sensitive and should be aligned with the pace and policy development in the right way. 

### 8.5. Setting for HIAs

The setting was an important part of the review aimed at defining how the specific concepts from the resources connect with the data provided. For this review, settings were not only limited to refer to the physical environment but also the intangible aspects and context in which HIAs were implemented in the health sector. There were eight projects, six proposals, two policies, and three programs that were subject to the HIAs. Using Table 2 and Table 3 below, it is easy to demonstrate how each concept is streamlined with the set parameters in the research work. 

Looking deeper at the projects, proposals, programs, and policies addressed in the HIAs mainly focused on either on the physical aspect or service configuration initiatives of the health sector.

Other minor themes identified were operational and workforce proposals which were addressed concurrently with the physical and service aspects of the HIAs. There were four service and operational oriented HIAs, one operational and workforce aspect oriented, one physical and operational oriented, and one combined service, operational, and workforce aspects oriented.

### 8.6. Sources of Evidence within HIAs

All the HIAs applied qualitative methods of data collection. Literature review was the most used method of data collection and was used in 13 of the HIAs [35,36,37,39,41,43,44,46,47,48,50,51,53] Interviews were conducted in eight of the included HIAs. Community-based focus group discussions were used in seven HIAs, while community and population profiles were used in five of the HIAs. The Consultation with the selected study participants, populations and experts in the relevant fields was carried out in five of the HIAs and previous studies, secondary data, and workshops were both reported in two HIAs. 

Other data collection methods that were used in a single publication each included: case studies, geographical information system analysis systems, questionnaires, previous cohort studies, meetings, professional knowledge, expert review, legislative analysis, geographical patches, cost-benefit analysis, and surveys. 

### 8.7. The Quality of Health Sector HIA Reports

The HIAs were appraised using the CASP qualitative checklist tool, using a set of 10 questions to consider the quality of the reports (Appendix A). These were rated as either ‘Yes’, ‘No’ or ‘Can’t tell’ by two reviewers, rating independently. Cohen’s K was used to determine if there was an agreement between the reviewers about the rating of the HIA reports’ quality, looking at the validity, detail, and applicability of research findings [34,54]. It is a platform that has to offer a common ground to deepen the score of communication as a reference point. According to Cohen, kappa values should be interpreted as ≤0 to mean no agreement, 0.01–0.20 to mean slight agreement, 0.21–0.40 to mean fair agreement, 0.41–0.60 to mean moderate agreement, 0.61–0.80 to mean substantial agreement, and 0.81–1.00 to indicate an almost perfect agreement [55]. There was substantial agreement between the authors’ ratings of the HIAs’ quality, based on the interpretation schema suggested by McHugh (2012), Ref. [55] with a calculated kappa of: κ = 0.629 (95% CI, 0.521–0.738). Therefore, there was good agreement between the two reviewers indicating the level of reliability in which the data collected are a true representation of the results [55]. 

## 9. Discussion

The health sector has utilized HIAs in policy making, program, projects, and proposal developments in a variety of diverse setting both physical and in service provision as reflected in the review. This is a clear indication that the scope of health sector practice and applications is broad, interdisciplinary, and complex [10]. However, the relatively small number of published reports available, 19 identified, on the health sector suggests there is currently an opportunity to increase and improve the use of the HIA tool in this sector. For instance, there is limited number of HIAs relating to clinical practice and clinical models of care irrespective of these areas being considered significant in health sector activity and expenditure [56]. 

On the other hand, the study shows different approaches can be utilized in the conduction of HIAs. Intermediate and rapid HIAs were the most prominent approaches in the health sector. This is highly consistent with other reviews which report higher rates in the conduction of rapid HIAs in comparison to other approaches namely intermediate, comprehensive, and desktop approach [10]. Factors considered in determining the approach implemented include timing, duration, scope, appropriateness, effort, complexity, and funds/resource available [10,40]. Rapid HIAs, as evident in the study, are quite prominently used since they require less time, often focus on less complex and smaller proposals, and primarily utilize literature review and qualitative analysis as their mode of data collection. HIAs are time-sensitive and should align with the pace of development and policymaking.

Considering the underlying values of HIA: equity, democracy, ethical use of evidence, and sustainable development [36], effective reporting and use of evidence should be emphasized especially in the health sector where evident based information is critical. The health sector is in most cases guided and supported by evidence-based approaches, informed by the evidence-based medicine movement more generally [57]. In contrast to this recommendation, however, the reports included in this review lacked in-depth details and documentation on each stage of the HIA process and justification as to why certain decisions were made. The stages involved in conducting HIAs include: screening, scoping, assessment, recommendations, reporting, monitoring, and evaluation [33]. The majority of the HIAs in this study do not include the six stages. It appears as though a flexible approach has been adopted for HIA practice to increase accessibility of the HIA tool in the studies identified (e.g., by allowing for the use of a wide range of data collection methods, the HIA opens itself up for use in a range of high and low resource settings). There is the possibility, for example, that by electing to use cheap and less time-consuming data collection methods (to save costs and time) HIA practitioners could miss out on valuable opportunities to learn and appreciate the value of more highly regarded data collection methods [37]. 

In addition, the lack of transparency around the scoping stage of the HIA process in the results also undermined the decision process resulting to the final choices around important factors like data collection methods and participants’ engagement. HIA guides recommend that during the scoping stage of the HIA, the decision-making process should be documented and in doing so justify their choice of HIA and associated methods [58]. 

Majority of the reports also lacked the monitoring and evaluation stage. Monitoring and evaluation are widely regarded as key elements of the HIA tool [11] and as Dannenberg and others argue, monitoring and evaluation should be continuously conducted throughout the HIA process to allow for the identification of unexpected issues and provides an opportunity for future improvements [11].

Finally, he identification of a single HIA conducted in the health sector in developing countries in contrast to eighteen in developed countries is an indication of an existing gap and opportunity for research and development. According to Erlanger et al.’s research, HIA practice is underutilized in developing countries [59]. It would be essential to determine the factors that have contributed to underutilization and how the use of HIAs can be encouraged and promoted in the health sector in these countries. 

## 10. Implications for Practice and Research

The underutilization of HIAs in the health sector globally and particularly in the developing countries indicates a growing need to further research and understand the barriers and hinderance limiting its use. Currently, the WHO identifies a lack of skills in research, critical appraisal, data synthesis, and analysis as one of the major barriers to HIA use [60]. As it stands, there is very little known about international health workers’ knowledge of HIAs. There are barely any studies that have been conducted to investigate the health sector professionals’ knowledge of HIAs. This limits the ability to effectively promote its usage and advocate for its advantages to inform decision-making. 

Countries could potentially investment in training programs in the health sector. Training programs for health workers could be paramount particularly in low-income settings or low resource countries, which are faced with limited funds to employ experts. This is another opportunity to identify means and ways to assist developing countries to build their capacity. Vulnerable communities should be prioritized in the implementation of HIAs. Dwyer (2004) calls for collaboration and restructuring of health systems and increased accountability from the government and community [61]. Availability of adequate funding and training would promote HIA practice.

As forementioned, HIAs play a critical role in decision making when implemented. However, shared decision-making in the health sector is highly dependent on clinical practice and models [62]. HIAs can therefore struggle to hold weight since health practitioners still prioritize traditional “scientific” biomedical type evidence of potential health impacts. Modern health care systems are complex [61] and decision making takes into account current best evidence, patient values, and clinical expertise [57,63]. One may assume, as a result, HIAs have seen relatively widespread use in other sectors as compared to the health sector [10]. For example, in land use planning, standards and benchmarks may be more influential in informing practice than the primary research and studies from which they’re derived. We cannot therefore just assume the health sector will incorporate the different approaches taken up by the other sectors. 

If HIAs are to be respected as rigorous tools for assessment and widely implemented in the health sector, they must be able to justify their use of the best available evidence. HIAs could potentially make better use of clinical data, particularly around monitoring and impact management processes. Improving and strengthening data collection and evidence through collaborative research would enable the health sector to better tackle health inequalities and foster equity [64]. On the other hand, ensuring the reporting standards are characterized by a fully documented and transparent process, clear statement of goals and aims, rigorous and documented data gathering and analysis, clear prediction of impacts, clear recommendations, and indicators to evaluate for quality and effectiveness of the HIAs would further promote its uptake [65,66]. The development of an HIA manual specifically designed to support health sector-based HIA report may present one solution to the current dearth of HIAs being conducted in the health sector. 

Other pathways in which HIAs can be implemented in the health sector involves addressing the social determinants of health, for instance, considering they play a critical role in influencing the population health [67]. On the other hand, an HIA tool could also address challenges that are hindering the achievement of Sustainable Development Goals. According to Kumar et al. (2016) Sustainable Development Goals (SDGs) are faced with challenges of accountability, high costs, maintenance of peace, and measuring of progress [68]. In addition, political commitment, the establishment of legal and policy frameworks, increased government or donor funding, and reallocation of resources in the health sector [69], would promote the use and implementation of HIAs in the health sector.

## 11. Limitations

The review included HIAs were subject to some limitations. The CASP tool (Critical Skills Appraisal Programme (CASP), 2006) appraisal highlighted opportunities to enhance the reporting of HIAs and suggest the need for further emphasis on identifying and discussing ethical concerns related to health proposals, policies, and programs. Major reporting elements were also lacking across the HIA reports reviewed. Moreover, majority of included material emerged from websites that were likely to include HIA reports. 

HIA Gateway [26], for instance, has historically been a notable and important repository for many HIA reports but has not received institutional support for several years and has not been updated. As such, the authors acknowledge that there will have been some HIAs conducted in the health sector that has not been published or hosted on these websites, for a range of reasons. The lack of representation of HIAs in developing countries in the review brings out the need to re-evaluate and identify factors attributing to this outcome. This can maybe be attributed to the lack of national policies mandating its use, and the high dependence on external funding and international lending agency compliance standards to prompt HIA’s use in developing countries [70]. Further to this, a lack of expertise and capacity in HIA has been identified in low-income countries [71]. 

Finally, Erlanger et al. argue that the reason a gap exists in the use of HIAs between high and lower-income countries is that in many developing countries, HIA is largely a slogan with limited practical implementation [59]. Despite the advantages of HIAs, it remains unknown who should be initiating an HIA. On top of that, it is very unclear at what stage, especially during policymaking should an HIA be implemented considering all the complexities involved. 

## 12. Conclusions

This review has highlighted the limited use of HIAs in the health sector. Health impact assessment is a tool that when implemented can help health sector actors to make better and more informed decisions regarding proposed health policies, programs, and projects as shown by the study. It enables allocation and reallocation of resources to the marginalized communities promoting equity and equality. If HIA’s use is to expand, it will be important for healthcare stakeholders and policy-makers to view HIA as a constructive and non-threatening process that usefully informs decision-making [40]. Exposing health professionals and health planners to HIAs’ use and enhancing reporting standards may assist in increasing its use. The use of HIA reviews is important in defining the quality-of-service delivery in varied settings. It will be possible to determine the course of use and usefulness of HIAs in the care setting as well as the community setting. Healthcare stakeholders will have an easy time in knowing exactly what is expected of them and the possible concern that is required in every level of interest in equating the services to foster efficiency and success of use. The use of HIAs places accountability onto the management of the hospitals hence increasing the level of efficiency in service delivery. 

## Figures and Tables

**Figure 1 ijerph-18-11466-f001:**
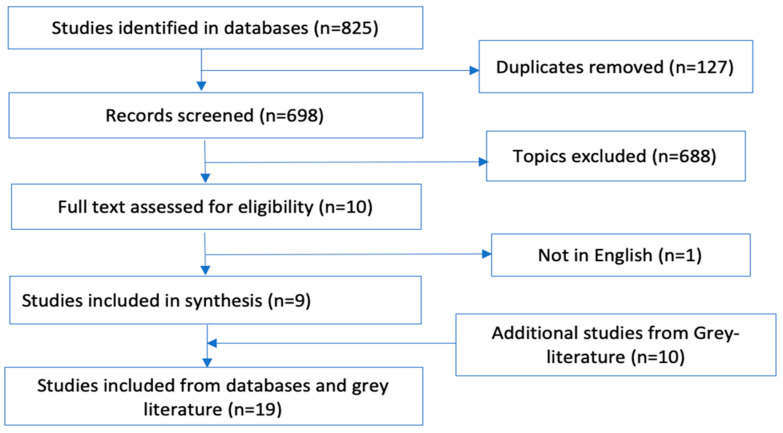
PRISMA summary flowchart of review of HIAs.

**Figure 2 ijerph-18-11466-f002:**
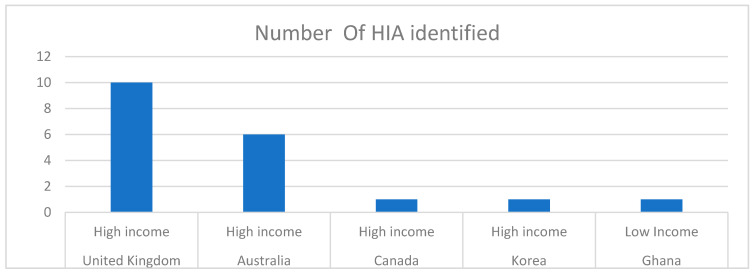
Distribution of HIAs.

**Table 1 ijerph-18-11466-t001:** Summary of included HIAs (*n* = 19).

Author and Title	Year	Country	Nature	Levels of HIA	Method	Objective	Outcome	Source
A health impact assessment on the construction phase of a major hospital redevelopment [35] (Maxwell et al.)	2008	Australia	Project-	Intermediate	-Literature Review -Consultation -Interviews	To provide a recommendation to hospital planners on how to minimize potential negative impacts and increase the positive impacts	Suggested creation of an asbestos removal strategy Recommended creation of a position that could facilitate the transition of the redevelopment process and management of certain requirements for the contractors’ contract. Example use of effective safety barrier	Database
Health impact assessment of free immunization program in Jinju City, Korea [36] (Kim et al.)	2012	Korea	Program-	Rapid	-Literature Review -Case Studies -Geographical information systems analysis -Questionnaire -Expert consultations	To evaluate the potential health impacts and identify the best way to improve the quality of the free immunizations by increasing gains and decreasing the risk	Separate recommendations were provided for both the private clinics and public health centres on reorientation of roles, management, development of better strategies, training, collaboration, and establishment of monitoring systems	Database
Learning from a Rapid Health Impact Assessment of a proposed maternity service reconfiguration in the English NHS [37] (Fell and Haroon.)	2008	United Kingdom	Proposal-	Rapid Desktop	-Published literature review -Cohort study of hospital study	To enable PCT and the Hospital Trust Board to evaluate the potential consequences of reconfigurations to access to maternal services, maternal and infant health amongst different socio-economic groups	The proposal showed the ability to reduce inequalities, which would be influenced by the saving attained through reconfiguration. Recommendations also ensured that the proposal went hand in hand with the government requirements with the implementation of a few strategies	Database
Health impact assessment for the sustainable futures of Salford [38] (Douglas et al.)	2004	United Kingdom	Project-	Comprehensive	-Interviews -Meetings -Published secondary sources	To provide quality services to patients and promote easy access to care and ensure personalized 24 h integrated health and social care.	Recommendations were made to the partnership between LIFT and SHIFT to explore and expand their operational definition of health. The community definition of health was wider than the medical and clinical perspective and was mainly based on healthy living and wellbeing	Database
A prospective mini health impact assessment of the ‘Towards 2010’ program in Sandwell and West Birmingham in the West Midlands [39] (Ali et al.)	2007	United Kingdom	Program-	Desktop Rapid	-Published literature -Professional knowledge	To identify and inform the Board on the most beneficial options of the program for the local population	Of the four evaluated options, two of them showed potential for maximum health benefits in health and wellbeing improvement Option A would improve existing facilities though has limited potential for a whole system approach Option B would improve both existing facilities and bring services closer to the community Options C and D had the highest potential for a whole system change with a broader linkage to regeneration initiatives	Database
A prospective health impact review of the redevelopment of Central Manchester Hospitals [40] (Bendel and Owen-Smith.)	2005	United Kingdom	Project-	Rapid	-In-depth discussions with key individuals	To ensure the social determinants of health had been considered. To identity and give recommendations for areas that needed to be improved to promote overall health impact To provide a pilot study that would be utilized across the Greater Manchester	Recommendations were made regarding the construction and design of the building, issues on the design and construction phase, and general issues that impact the wider community. Example of these issues included parks, riding facilities, housing, crime, and extreme weather conditions	Database
A health impact assessment of the Liverpool Hospital redevelopment [41] (Maxwell.)	2007	Australia	Project-	Intermediate	-Literature Review -Key Interviews -Population Profile	To evaluate the health impacts of the project on the community, patients, and their families	The recommendations were made in terms of the priority of the health impacts. The issues were prioritized as follows: reduced parking, health and wellbeing, community and patient safety and increased traffic,	Database/Grey literature
Health Impact Assessment of the Northern Territory Emergency Response [42] (Australian Indigenous Doctor’s Association and Centre for Health Equity Training)	2010	Australia	Project-	Comprehensive	-Legislative analysis -Community consultations -Community profile -Expert review	To identify potential positive, negative, and unpredicted health impacts of the Northern Territory Emergency Response	The issues identified to have potential health impacts included: external leadership, Education, governance and control, housing, prohibited materials, children health, and income and alcohol restrictions Recommendations were made to either be stopped, unlikely potential to be effective in the long run, or the option to proceed with caution	Grey literature
The Health Impact Assessment Statement on the NSW Integrated Chronic Disease Prevention Campaign [43] (O’Hara B.)	2004	Australia	Proposal-	Intermediate	-Community profile -Literature review and Exploration of themes -Expert opinion and key informant interviews -Focus group discussions	To provide information that could be utilized to better the proposal and enhance the positive impacts while at the same time decreasing the negative impacts	Recommendations were made relating to various components of the campaign: mass media communication, branding, public relations, community support, promotion of appropriate referral point and appropriate, accessible information and merchandise	Grey literature
West-Stockwell Primary Health and Community Resource Centre A Health Impact Assessment [44] (Cooke and Bowman.)	2004	United Kingdom	Project-	Rapid	-Community-based sessions -Interviews -Published literature	To provide recommendations on how to mitigate short term negative impacts on health and assist West Stockwell Partners Board in decision making	The recommendations were made in terms of service provision, safety, communication, communication development, community space provision, and means of easily manoeuvring around	Grey literature
Rapid Health Impact Assessment of the Private Finance Initiative Proposal “Modernising Healthcare for a deprived Community” [45] (Ardem.)	2003	United Kingdom	Proposal-	Rapid	Unclear	To measure potential health outcomes of the proposal against social gains and sustainable financial support	Recommendations were made for readjustment and modification in the development phases to improve the potential health outcome	Grey literature
A Rapid Health Impact Assessment of “Our Health, Our Care, Our Say” on Young Carers [46] (Abrahams and Pennington.)	2008	United Kingdom	Policy-	Rapid	-Secondary sources -Published literature -Interviews -Focus groups	To inform decision making during the policy planning process by illustrating the health implications of the policy	The policy illustrated potential opportunities to promote systems improvements, increase available resources and support the younger population and their families. On the other hand, it opened a chance for the introduction of drivers promoting the status of young carers and opportunities to increase their visibility across health and social care services through various recommendations made	Grey literature
Report of the Health Impact Assessment of Proposed Changes to Mental Health Services [47] (Shepherd.)	2011	United Kingdom	Proposal-	Rapid	-Consultations -Literature review -Workshops	To inform the Board on the strengths and weaknesses of their proposal to make reforms and aid in decision making	Results showed the presence of an opportunity for the Board to develop a good relationship between itself and the community to address the various concerns presented and those that may emerge such as stigmatization and assumptions made to maximize health benefits.	Grey literature
Future Directions for Health Promotion. 2008, Mid-Western Health Service [48] (NSW Health)	2008	Australia	Proposal-	Rapid	-Literature review -Geographic patches -Surveys -Interviews	To identify whether the changes would have any positive impact or unpredictable consequences to the communities in MWAHS and offer recommendations for improvement.	Strategic approach recommendations were made focusing on the development of the workforce, prioritizing plans and policy on equity issues, targeting the vulnerable communities, and ensuring a supportive infrastructure at the local level	Grey literature
Report on Rapid Health Impact Assessment of CADMHAS [49] (Critchley et al.)	2007	United Kingdom	Program-	Rapid	-Participatory workshop	To ensure that the positive health benefits were maximized, and the negative impacts are highly decreased for the health and wellbeing of the population	Recommendations to improve health and wellbeing and reduce the inequalities within the community through CADMHAS were made through service delivery.The implementation was highly dependent on the amount of funding the proposal received.	Grey literature
Reviewing a Rural Health service Redesign Proposal Using the Health Impact Assessment (HIA) Process [50] (Neumayer.)	2004	Australia	Proposal-	Rapid	-Community profile -Literature review -Scenario Approach -Cost–benefit analysis	To identify the consequences of the proposal in terms of quality of service, access, availability, and the workforce	Recommendations were made to support the reconfiguration of the existing services against the model of no change to current existing services	Grey literature
The Dulwich WellBeing Centre… a Health Impact Assessment [51] (Atkinson et al.)	2003	United Kingdom	Project-	Rapid	-Literature review -Community profile -Community-based discussions -Workshops -Previous research conducted as preliminary thinking of the project -Mapping of older people’s services	To provide information that could be utilized to strengthen and improve health opportunities and capture the views of the community and their support	Both short term and long-term recommendations were identified since the results indicated the need for a more holistic model which embraces the concept of wellbeing. The centre would provide a sense of ownership and identity for the community other than just delivery of services or a physical space	Grey literature
Health Impact Assessment, Municipal Development Practices, And Children’s Health [52] (Rattle.)	2015	Canada	Project-	Rapid Desk-based	-Limited discussion	To identify and evaluate the impacts of the project, to recommend ways to mitigate the impacts, and to inform future policies makers and program development of any similar nature that may be undertaken	Different populations were affected differently. The immediate community was advantaged from easy access to health services thus improved health. On the other hand, residents, those attending school and work around the area would be faced with increased traffic, air pollution, noise, decreased green space as well as stress.	Database
Community Health Impact Assessment in Ghana: Contemporary Concepts and practical methods [53]	2019	Ghana	Policy	Desktop	-Focus group discussion, -Key informant interviews -literature review.	To identify the impact and influence of the NHIS policy on health and the type of influence on the population.	Not discussed. Article focused and discussed processes and methods utilized in the HIA.	Database

**Table 2 ijerph-18-11466-t002:** Setting for HIAs.

Setting for HIAs	Number of HIAs	Definition of Terms and Examples
Projects	8	Project in the HIAs was used to refer to a specific activity that was undertaken in a specified location. Projects included focused on issues such as a construction phase of Manchester hospital in the United Kingdom [40], redevelopment of Liverpool Hospital in Australia [41] and assessment of the Northern Territory Emergency [42].
Proposals	6	Proposals referred to all formal plans that were put forward for consideration for future implementation [13]. Proposals focused on redesigning rural health services, health promotions [48] and changes to mental health services [47] amongst others.
Programs	3	A program was described as a set of related activities that eventually give effect to the policy [13]. The programs reviewed addressed issues such as free immunization in Korea and chronic disease prevention campaign in Australia [36,43].
Policies	2	Policy, contrary to proposals and projects referred to how an organization or the government aims to achieve a set of objectives. The policy subject to HIA conducted in the United Kingdom dealt with the health of the young populations and opportunities available for improvements of their health and that of their families [46] while the one in Ghana focused on Ghana’s National Health Insurance Scheme [53].

**Table 3 ijerph-18-11466-t003:** Focus of HIAs.

Number of HIAs	Focus of HIAs	Examples
12	Service and service reconfiguration	The services and service reconfiguration aspects included: free immunization, maternity service, chronic disease prevention, primary health care, health care for vulnerable communities, youth and young carers service, mental health, mental health advocacy, national health insurance scheme, and rural health service reconfiguration of the HIAs [36,37,39,42,43,46,47,48,49,50,52,53].
7	Physical	The HIAs that assessed proposed physical changes to the health system focused on reconstruction, redevelopment, and development of hospitals about the HIAs [35,38,40,41,44,45,51].

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
