# Peer review of "Health Impact Assessments of Health Sector Proposals: An Audit and Narrative Synthesis"

_ijerph, 2021, doi:10.3390/ijerph182111466_

Round 1

Reviewer 1 Report

Minor editing issues exist such as spacing in Abstract (lines 18, 28 etc.) and needed correction in lines 99 and 323. Recommend to check throughout the manuscript before its publication.

In lines 120-123, examples of keywords, synonyms, and MESH terms need to be provided even though the authors provided them in Supplementary file 1. Plus some readers are not familiar with the MESH terms. So explain the MESH terms in the text.

In lines 171-172, more detailed information about how the 689 were excluded needs to be provided in the text.

In lines 181-183, more explanations and implications of the levels of HIAs need to be provided in the text.

In line 261, provide implications of underutilized HIA practice in developing countries

In lines 263-268, detailed examples need to be provided in the text

Most contents in the sections of the discussion and the implications for practice and research are felt like general findings. The authors need to highlight their own research findings more clearly.

Author Response

Dear Reviewer 1,
Thank you for taking time to review our manuscript. We really appreciate the valuable feedback you have provided.
Please see below response to the comments provided:
1. Minor editing issues exist such as spacing in Abstract (lines 18, 28 etc.) and needed correction in lines 99 and 323. Recommend to check throughout the manuscript before its publication. 

Manuscript reviewed and editing issues addressed.

2. In lines 120-123, examples of keywords, synonyms, and MESH terms need to be provided even though the authors provided them in Supplementary file 1. Plus some readers are not familiar with the MESH terms. So explain the MESH terms in the text.

Addressed in line 192-194

3. In lines 171-172, more detailed information about how the 689 were excluded needs to be provided in the text.

Addressed in line 249-255

4. In lines 181-183, more explanations and implications of the levels of HIAs need to be provided in the text.

Addressed in line 352-361
5. In line 261, provide implications of underutilized HIA practice in developing countries –

Addressed in the manuscript implication section. (line 463-470)
6. In lines 263-268, detailed examples need to be provided in the text. 

Addressed in line 352-361

7. Most contents in the sections of the discussion and the implications for practice and research are felt like general findings. The authors need to highlight their own research findings more clearly.

Edited to be more research finding specific and focused.
Kind Regards,
Nelius Wanjohi.

Reviewer 2 Report

  1. Abstract is lengthy, and should be simplified and structured.
  2. There are only 19 materials for HIAs founded in this research, why is the reason?  Researchers have to clarify the importance of HIAs, there are so many other systems adopted by the health domain.

Author Response

Dear Reviewer 2,
Thank you for taking time to review our manuscript. We really appreciate the valuable feedback you have provided.
Please see below response to the comments provided:
1. There are only 19 materials for HIAs founded in this research, why is the reason?

The systematic search process identified 19 reports eligible for inclusion relevant to the inclusion criteria for this review. The eligibility criteria has been discussed in line 171 to 188.

2. Researchers have to clarify the importance of HIAs, there are so many other systems adopted by the health domain.
The review focused on the Health Impact Assessment tool use in the health sector due to its significance and widespread utilisation/implementation in intersectoral activity to promote health, in contrast to its relatively limited use within the health sector itself. This justification has been clarified in the introduction.
Kind Regards,
Nelius Wanjohi.
